# From Childhood Woes to Adult Blues: Unmasking the Role of Early Traumas, P2X7 Receptor, and Neuroinflammation in Anxiety and Depression

**DOI:** 10.3390/ijms26104687

**Published:** 2025-05-14

**Authors:** Zsuliet Kristof, Dorottya Szabo, Beata Sperlagh, Dora Torok, Xenia Gonda

**Affiliations:** 1Discipline of Psychiatry, Adelaide Medical School, The University of Adelaide, Adelaide, SA 5000, Australia; zsuliet.kristof@adelaide.edu.au; 2Laboratory of Molecular Pharmacology, HUN-REN Institute of Experimental Medicine, 1083 Budapest, Hungary; szabo.dorottya@koki.hun-ren.hu (D.S.); sperlagh@koki.hu (B.S.); 3Janos Szentagothai Doctoral School, Semmelweis University, 1085 Budapest, Hungary; 4Department of Pharmacodynamics, Semmelweis University, 1089 Budapest, Hungary; torok.dora@semmelweis.hu; 5Center of Pharmacology and Drug Research & Development, Semmelweis University, 1089 Budapest, Hungary; 6Hungarian Brain Research Program, NAP3.0-SE Neuropsychopharmacology Research Group, 1089 Budapest, Hungary; 7Department of Psychiatry and Psychotherapy, Semmelweis University, 1082 Budapest, Hungary; 8Department of Clinical Psychology, Semmelweis University, 1092 Budapest, Hungary

**Keywords:** P2X7R, early-life stress, anxiety, depression, neuroinflammation, childhood trauma

## Abstract

Early-life stress may increase the risk of neuropsychiatric disorders via immune activation. While the purinergic signaling pathway is implicated in psychiatric disorders, the specific role of the P2X7 receptor (P2X7R) in anxiety, depression, and childhood trauma still requires further clarification. Upon chronic stress, excessive ATP release activates purinergic P2X7R signalling in the brain contributing to long-lasting neuroinflammation, which potentially promotes the development of psychiatric disorders. There is also a putative link between the P2X7 receptor gene, located on chromosome 12q24, and the development of anxiety and depression. This review aims to systematically examine how P2X7R contributes to the pathophysiology of anxiety and depressive disorders, with a particular focus on early-life stress (ELS). It offers a comprehensive synthesis of the current findings, emphasizing the previously unexplored intersections between P2X7R signaling, early-life stress, and psychiatric disorders. These interactions may shape long-term neuroinflammation, contributing to the development of anxiety and depression, and offer new insights into potential therapeutic targets. The review integrates the role of P2X7R regarding both indirect mechanisms—such as the modulation and long-term transmission of neuroinflammation following environmental stressors and vulnerability—and direct genetic associations with psychiatric conditions, including the influence of single-nucleotide polymorphisms (SNPs), haplotypes, and other variants within the P2X7 gene. Special emphasis is placed on the impact of early-life stress, drawing primarily on preclinical findings to elucidate underlying mechanisms.

## 1. Introduction

Stress is a persistent life experience that impacts physiological and emotional responses, cognitive function, and daily behaviours. A growing body of evidence shows that childhood exposure to major adverse experiences, such as abuse, neglect, and parental loss or substance abuse, disrupts both physiological and psychological functioning, enhancing vulnerability to neuropsychiatric disorders, particularly depression and anxiety [1], and physical illnesses, such as coronary heart disease, diabetes, or autoimmune conditions, across the lifespan [2]. Stressful events occurring during this critical developmental period not only seem to lead to behavioural and psychological difficulties later in life, but they also appear to alter immune system functioning, resulting in chronic immune activation [1]. This chronic inflammatory state in the brain is thought to contribute to the pathogenesis of various neuropsychiatric disorders, including depression, anxiety, ADHD, and autism spectrum disorder [3,4]. Furthermore, emerging evidence suggests a link between early-life trauma, heightened inflammatory responses in adolescence and adulthood, and subsequent behavioural and cognitive alterations [1].

Microglia, the resident immune cells of the central nervous system (CNS), play an essential role in neuroinflammation and are involved in the brain’s defence responses to adverse stimuli, as well as in neuronal activity and synaptic plasticity [5]. These macrophage-like cells are sensitive to environmental changes in the brain and play a crucial role in modulating the effects of stress [6]. Chronic stress-induced microglial activation triggers the release of proinflammatory interleukin (IL)-1β, IL-6, and tumour necrosis factor (TNF)-α in the limbic system and prefrontal cortex, besides further modulators, such as glutamate, chemokines, nitric oxide, and growth factors. These inflammatory mediators disrupt key processes like neurotransmission, neuronal plasticity, and neurotrophic factor signalling, contributing to the development of psychiatric disorders [7,8,9].

A key mediator of microglia-induced neuroinflammation is the purinergic P2X7 receptor, which is a non-selective cation channel abundantly expressed in microglia [10]. The P2X7 receptor belongs to the P2X receptor family, which are ionotropic ATP-gated receptors formed by three subunits of the P2X1-7 subtypes. These trimeric receptors open in response to ATP binding, allowing calcium and sodium influx and potassium efflux, thereby influencing membrane potential and downstream signalling [11]. Among the P2X family, P2X7R stands out due to its unusually long C-terminal tail, distinct ability to form a large non-selective pore upon sustained activation, and markedly low affinity for ATP (0.1–1 mM), making it selectively responsive under pathological conditions. These structural and functional features underlie its central role in inflammation-associated cell death and are further detailed in Section 3.1.

Mounting research indicates that the abnormal expression and activity of mammalian P2X receptors contribute to the development of several diseases, particularly those involving neuroinflammation [12]. Recent investigations have identified a novel homozygous deletion in the *P2RX6* gene, likely disrupting protein function and implicating it in the aetiology of myopathy [13]. Additionally, *P2RX4* has been proposed as a regulatory target of the macroautophagy/autophagy–lysosome pathway, suggesting a potential role in the pathophysiology of Parkinson’s disease [14]. Of the P2X family, *P2RX7* is the most extensively studied and has been implicated in a range of pathological conditions, including arthritis [15], chronic pain [16], cardiovascular diseases [17], and cancer [18,19]. The human *P2RX7* gene contains numerous single-nucleotide polymorphisms, including many non-synonymous variants (NS-SNPs) that lead to changes in the receptor’s amino acid sequence. Similarly, the mouse *P2RX7* gene also features NS-SNPs. Studies characterizing these naturally occurring P2X7 variants have revealed new functional insights and associated specific mutations with an increased risk for certain diseases [12,18]. In recent years, large-scale screening efforts have uncovered a wide array of selective and structurally varied P2X7R antagonists, some of which exhibit favourable drug-like characteristics and demonstrate therapeutic benefits in rodent models [20]. Building on its well-established role in peripheral inflammatory conditions, there has been a recent shift in the literature toward exploring the involvement of P2X7R in CNS-related disorders, including depression [21], anxiety [4], multiple sclerosis [22], Alzheimer’s disease [23], epilepsy [24], schizophrenia [25], and autism spectrum disorder [26], further highlighting its broad pathological relevance.

The ion channel can be activated by a stress-induced sharp elevation in extracellular ATP. The downstream effects of P2X7R activation extend beyond NLRP3-mediated pro-inflammatory IL-1β and IL-18 release [10,26,27,28,29]; the receptor is also implicated in various processes associated with depression and anxiety, including disrupted monoaminergic neurotransmission, heightened glutamatergic transmission, and reduced neuroplasticity [30]. Additionally, P2X7R directly regulates hippocampal glutamate release and influences the brain-derived neurotrophic factor (BDNF) signalling pathway, which are crucial for neuronal plasticity and are often disrupted in mood disorders [31,32].

The potential role of P2X7R in these disorders is also reflected in the fact that its activity and hippocampal expression are selectively regulated by exposure to various antidepressants and stress [30]. The blockade of the P2X7-NLRP3-IL1β pathway may promote stress resilience via the involvement of microglia and monocytes [33], thus supporting the potential role of P2X7 signalling as a connecting point between chronic stress and clinical mood symptoms [34,35,36]. Therefore, in this review, we discuss the current evidence linking P2X7R to stress, depression, and anxiety by overviewing the most recent findings from both human and animal research to deliver an integrative view of its supposed underlying neurochemical and molecular mechanisms.

## 2. Early-Life Trauma and Neuroinflammation: Setting the Stage

### 2.1. The Potential Role of Neuroinflammation in the Lasting Effects of Early-Life Traumas

Multiple strands of evidence suggest that immune dysfunction plays a role in the manifestation of a pro-inflammatory profile in the pathophysiology of depression and anxiety [37,38].

It is known that early-life stress provokes neuroimmune responses that might lead to prolonged, pathological, and maladaptive neuroinflammation [1]. Children and adolescents who experience chronic psychological stress show increased levels of peripheral inflammatory factors, in comparison with their non-exposed peers. Emerging evidence suggests that these effects can be long-lasting, leaving an inflammatory residue [39]. Recent meta-analyses found significant associations between childhood maltreatment and inflammatory markers in adulthood, with generally increased peripheral levels of TNF-α, IL-6, and CRP in adults and elevated concentrations of TNF-α only in the case of sexual abuse victims [40]. Furthermore, in adolescent women experiencing early-life abuse (ELA), increased IL-6 concentrations predicted the onset of depression 6 months later [41]. These associations seem to be independent of the adult’s concurrent psychosocial and socioeconomic conditions, underpinning that childhood adversity leaves an inflammatory mark [42].

### 2.2. The Role of Early-Life Stress in Shaping Neuroimmune Responses

Animal models investigating early adversity utilize stress exposure in utero or during early postnatal time windows, which can induce lasting alterations in mood-related behaviour, remaining long after the ceasing of stress [43,44]. In the postnatal period, most rodent models of ELS apply the perturbation of dam–pup interactions and derangement of the dam’s caregiving behaviour [45], which commonly disturbs stress-responsive neuroendocrine pathways, drives neuroinflammatory states, and evokes epigenetic changes. These processes result in structural and functional alterations in neurocircuits regulating anxio-depressive behaviours, such as the hippocampus, prefrontal cortex, amygdala, and the brain-stem monoaminergic nuclei, leading to enhanced anxio-depressive behaviours in the offspring, often accompanied by disturbed cognition, reward, and social behaviour [46,47]. Exposure to brief daily separation evokes similar clinical features as observed in maltreated children, such as increased CRP and IL-6 [42,48,49,50], reduced myelination, and elevated anxiety-like behaviours during the juvenile period and adulthood [51,52,53,54].

Examining the short- and long-term consequences of maternal separation (MS) in animal models, Zhou et al. (2020) revealed different behavioural responses in adulthood to stress exposure, after various degrees of separation during early life. While long-term MS induced stress sensitivity, activated neuroinflammation, and impaired neuroprotection, short-term MS improved neuroprotection to gain stress resistance in adulthood [55]. Encountering stress during the postnatal period results in heightened microglial density and activity within the hippocampus, which is associated with modifications in the regulation of various developmental genes involved in cell cycle progression, inflammation, and cell migration [51,56].

Extensive research indicates that disruptions in microglial function during specific developmental phases lead to enduring structural modifications and behavioural changes in psychiatrically relevant domains such as anxiety, depression, and social affiliation, persisting into adulthood [51,57,58,59,60].

Altered cytokine concentrations might be a result of gene polymorphisms, which could contribute to later sensitivity to other factors, such as early childhood maltreatment [61]. As the recognition of the significant role of neuroinflammation in comprehending neuropsychiatric disorders and associated symptoms continues to expand, the P2X7 receptor is emerging as a focal point of considerable interest as a potential central hub in brain-related disorders [62]. However, to date, studies investigating the role of the P2X7 receptor and gene in anxiety and depression are scarce, and only a few considered the role of psychosocial stress, the main environmental risk factor for these disorders.

## 3. P2X7 Receptor: A Central Hub in Neuroinflammation

### 3.1. Role of P2X7R in ATP Signalling

The P2X7 receptor has a fundamental role in maintaining and amplifying inflammation and may contribute to the development of long-term inflammatory states. ATP is stored in vesicles and can be released from nerve terminals, dendrites and axons [63], astrocytes [64], and microglia. Under normal neuronal activity, ATP acts as a local messenger, facilitating intercellular communication and maintaining homeostasis. Its release and the subsequent activation of P2X7 receptors on neighbouring cells contribute to this balanced state [65]. In inflammatory conditions, substantial increases in extracellular ATP arise from both active release and passive leakage from damaged and apoptotic cells [66].

Unlike other P2X receptors, P2X7R has a lower affinity for ATP (requiring concentrations between 0.1 and 1 mM for activation). Consequently, P2X7R activation primarily occurs under pathological conditions where extracellular ATP levels are elevated, such as during inflammation, cellular trauma, hypoxia, or other adverse events that cause cell death and ATP release [67,68]. Prolonged P2X7R activation by high extracellular ATP leads to rapid ion fluxes (Ca^2+^ and Na^+^ influx, K^+^ efflux), forming a large non-selective membrane pore permeable to molecules up to ~900 Da, ultimately causing cell death (lysis, necrosis, or apoptosis) [69]. The structural basis of these effects lies in the receptor’s architecture: P2X7R is a homotrimer, with each subunit containing two transmembrane domains (TM1 and TM2), a large extracellular ATP-binding loop, and intracellular N- and C-terminal tails. The TM2 domain lines the ion-conducting pore and is critical for channel gating, while the unusually long intracellular C-terminal tail—distinctive to P2X7R—regulates downstream signalling events, including NLRP3 inflammasome activation and apoptosis-related pathways [11]. These unique structural features support the receptor’s dual role as both an ion channel and a pro-inflammatory trigger under stress conditions. The structural organisation of the P2X7 receptor, including its transmembrane domains and intracellular signalling regions, is illustrated in Figure 1.

### 3.2. The Role of P2X7R in Neuroinflammatory Processes

P2X7R is a multifaceted receptor that participates in a wide range of physiological and pathological conditions in different body systems by modulating cellular responses in both immune and non-immune cells. Its effects might be different and contrasting, and are highly context-dependent varying with the level of receptor activation, cell type, type of pathogen, and severity of disease. In some conditions, particularly within infectious inflammatory disease and cancer, P2X7R may act as either a protective or harmful factor [70]. In the innate immune system, P2X7R activation via extracellular ATP—released in response to pathogen- or damage-associated molecular patterns—triggers pro-inflammatory cascades that constitute the first line of defence against pathogens [71]. Moreover, its role extends beyond the innate immune response. P2X7R influences adaptive immune response by regulating T-cell signalling and activation, polarization, metabolism, memory formation, tissue residency, and susceptibility to cell death, thereby playing an essential role in many processes regulating immune system homeostasis and response [70,72].

Interestingly, P2X7R also contributes to non-inflammatory immune functions. It facilitates the recognition and phagocytosis of foreign particles and apoptotic cells in the absence of extracellular ATP or opsonins, indicating that it participates in the silent clearance of cellular debris—an essential process for avoiding chronic inflammation and autoimmunity [73]. Together, these findings illustrate that P2X7R signalling is not inherently detrimental; rather, it plays a nuanced role that can be beneficial depending on the context. Its function spans both immune activation and resolution, suggesting that total inhibition may not always be desirable and could potentially disrupt essential protective or regulatory mechanisms.

Toll-like receptors (TLRs), innate immune receptors found in immune, glial, and neuronal cells, are involved in both infectious and non-infectious CNS processes and play a pivotal role in mediating inflammatory responses [74]. TLR signalling must be strictly regulated to maintain immune homeostasis [75]. ELS can prime the immune system and heighten the sensitivity of TLR signalling pathways [76]. This alteration induced by ELS in immune responsiveness can trigger excessive neuroinflammatory responses upon subsequent environmental or psychological stressors, leading to a detrimental cycle of inflammation with chronic inflammatory conditions [75,76,77,78,79]. The pro-inflammatory environment can impair neuroplasticity and contribute to long-term behavioural abnormalities [75]. This involves the NF-κB-dependent transcription of cytokine precursors, followed by P2X7R activation, NLRP3 inflammasome assembly, caspase-1 activation, and ultimately, the release of mature IL-1β [80,81]. This amplifies the inflammatory response and contributes to long-term behavioural abnormalities [82]. The interplay between ELS and TLR-mediated neuroinflammation underscores the need for a comprehensive understanding of the mechanisms behind chronic stress-related disorders. The dysregulation of TLR signalling not only establishes a foundation for chronic neuroinflammation but also exemplifies how early adversities can imprint onto biological systems.

Microglia, the resident immune cells of the CNS, are known to widely express P2X7 receptors on their surface, which play a crucial role in mediating inflammatory responses and modulating neuroinflammation [83]. The chronic or excessive activation of microglia through P2X7R results in the previously described NLRP3-mediated neuroinflammation, characterized by increased cytokine levels, reactive oxygen and nitrogen species, proteases, and excitotoxic glutamate [25,83,84], which can impair neuroplasticity, alter neurotransmission (particularly glutamatergic signalling), or directly damage neurons, contributing to behavioural and morphological changes [25,85]. Beyond inflammasome activation, P2X7R also influences other microglial functions, such as phagocytosis, the process by which microglia engulf and remove cellular debris and pathogens [86]. These P2X7R-mediated microglial effects have been implicated in the pathophysiology of several psychiatric disorders, including schizophrenia, depression, and anxiety [25]. Preclinical studies show that P2X7 receptor knockout in microglia prevents stress-induced depressive and anxiety-like behaviours, while P2X7R agonists induce such behaviours, highlighting microglial P2X7R as a potential therapeutic target for depression and anxiety [83]. The molecular cascade linking early-life stress, TLR sensitization, ATP-driven P2X7R activation, and inflammasome-mediated cytokine release in microglia is summarized in Figure 2.

### 3.3. P2X7 Receptor Distribution and Function in the CNS

Functional P2X7 receptors are widely expressed in various tissues. In the CNS, they are found in areas relevant to the generation and development of anxiety and depression, such as the frontal cortex, amygdala, hippocampus, and striatum. According to Human Protein Atlas [87,88], the mRNA expression levels of the *P2RX7* gene showed high expression in white matter, basal ganglia, pons, medulla oblongata, and thalamus (Figure 3).

These receptors are also expressed in diverse cell types, including microglia [89], oligodendrocytes (Figure 4), and Schwann cells [90], while their expression and functionality in neurons and astrocytes remain debated [33,91].

The P2X7 receptor plays a role in processes related to neuroinflammatory response, impairment of neuroplasticity, and the stimulation of glutamate response [62]. P2X7R influences cellular proliferation and death, rapid as well as reversible phosphatidylserine exposure, membrane blebbing, microparticle and exosome release, multinucleated cell formation, and reactive oxygen and nitrogen species production [62].

### 3.4. P2X7R’s Involvement in Other Neurological and Psychiatric Conditions

P2X7R is implicated in many CNS disorders, such as multiple sclerosis, Alzheimer’s disease, Parkinson’s disease, epilepsy, chronic pain, and psychiatric disorders such as major depression (MDD), bipolar disorder (BD), schizophrenia, and autism spectrum disorder [25,93]. Comorbidity between central nervous system diseases and peripheral inflammatory disorders likely arises from a compromised immune system. This allows peripheral immune cells and signalling molecules to interact with the CNS, both directly (via blood–brain barrier disruption) and indirectly (through meningeal immunity), combined with inherent CNS neuroinflammation driven by microglia [94]. Targeting P2X7R with antagonists appears to counteract neuroinflammation in preclinical models [93,95]. Therefore, brain-penetrant P2X7R antagonists hold potential as therapeutics for neuropsychiatric disorders [93,96]. However, this needs further validation through clinical data.

## 4. P2X7R Activation: A Bridge Between Early Trauma and Anxio-Depressive Behaviours

### 4.1. Early-Life Stress Leads to Prolonged Enhanced Immune Activation

Cumulative mild stress during early life promotes long-lasting anxiety and depression-like behaviours in young adult mice along with increased inflammatory cytokines, especially interleukin (IL)-17; upregulated microglial activation in the hippocampus, amygdala, and prefrontal cortex; and changes in the T-helper (Th)-17 cell population; as well as differentiation in mouse brain samples [97]. Upon acute stress, the HPA system releases glucocorticoids that stimulate both the well-known anti- and lesser-known pro-inflammatory responses [98,99] (Figure 1). Acute stressors increase the release of glucocorticoids to reduce inflammation [98]. Over time, chronic stress shifts the response to pro-inflammatory, allowing the organism to respond to sustained threats [99] through glucocorticoid-dependent NLRP3 induction, which sensitizes the cells to extracellular ATP and significantly enhances the ATP-mediated release of proinflammatory molecules, including mature IL-1β, TNF-α, and IL-6 [100] (Figure 1 and Figure 5). While the exact mechanisms by which early-life stress leads to P2X7R upregulation remain unclear, it might upregulate P2X7R through multiple interacting pathways: epigenetic changes to the *P2RX7* gene itself, elevated glucocorticoid signalling impacting *P2RX7* transcription or microglial activity, neuroinflammation creating a positive feedback loop with IL-1β further increasing P2X7R expression, and increased ATP release and subsequent P2X7R sensitization or microglial priming leading to enhanced ATP responses (Figure 5). Emerging evidence positions the *P2RX7* as a key mediator at the intersection of neuroinflammation and stress-related psychiatric conditions, including depression and anxiety. A brief overview of pathway-level information highlights the mechanistic relevance of *P2RX7* as a molecular hub mediating neuroinflammatory responses that may contribute to the onset and progression of psychiatric disorders, by linking early-life stress, inflammation, and affective dysregulation. According to QuickGO annotations (https://www.ebi.ac.uk/QuickGO/annotations?geneProductId=Q99572, accessed on 6 May 2025), *P2RX7* is implicated in several key biological processes and pathways. Gene Ontology (GO) terms highlight its involvement in the inflammatory response, regulation of interleukin-1β production, and positive regulation of reactive oxygen species metabolic process, all of which are central to neuroimmune activation. Furthermore, pathway mapping via the KEGG database (https://www.kegg.jp/kegg-bin/search_pathway_text?map=map&keyword=p2rx7&mode=1&viewImage=true, accessed on 6 May 2025) reveals *P2RX7*’s participation in the NOD-like receptor signalling pathway, neuroactive ligand-receptor interaction, and the calcium signalling pathway. These pathways are known to mediate cellular stress responses and neuroinflammatory cascades that can be sensitized by early-life stress exposures. The chronic activation of these mechanisms has been implicated in long-term changes to brain circuitry and behaviour, aligning with current models of the neurobiological embedding of early stress into psychiatric vulnerability.

### 4.2. Neuroinflammation and Disruption of Brain Development

Neuroinflammation initiates microglial activation. These cells, integral to cerebral inflammatory responses and neuronal pruning, express glucocorticoid receptors [101]. These receptors are highly expressed in the prefrontal cortex and hippocampus during key developmental periods [99,102]. Inflammation can also directly influence GABAergic/parvalbumin neurons, thereby altering the excitatory/inhibitory balance and intensifying the impact of stress on regional brain development [99]. Thus, the interaction of stress that occurs during a sensitive period of development, in response to the effects of inflammation, produces a permanent change in neural functioning. For example, this can lead to atrophy of pyramidal neurons in the hippocampus and depletion of synaptic vesicles, as observed in an animal study [103].

### 4.3. Early-Life Immune Activation Leads to Long-Term Neuropsychiatric Disorders

The early-life stage is a critical period for neurodevelopment, where immune system disruptions can lead to lasting consequences, increasing the risk of neuropsychiatric disorders [104] (Figure 5). A mouse model of allergic dermatitis in the early-childhood period revealed that ELS due to allergic dermatitis impacts mental health in adolescence by causing microglia to be in a priming state for later stressful events, which results in high susceptibility to systemic inflammation in adolescent animals, coupled with an elevated expression of pro-inflammatory cytokines and heightened proliferative capacity of microglia, leading to depressive behaviours [105]. Another animal study highlights the significant role of the P2X7 receptor in the inflammatory processes associated with irritant contact dermatitis, suggesting a potential therapeutic approach by targeting P2X7R leading to the suppression of immune cell infiltration and cytokine release [106]. A chronic inflammatory skin disease with childhood-onset, atopic dermatitis (AD) not only causes significant stress conditions [107], but has also been linked with psychiatric disorders, including depression, anxiety, ADHD, and autism [108]. These findings underscore the significant mediatory role of the P2X7 receptor in bridging ELS with long-term mental health outcomes. Beyond its involvement in skin-related conditions, P2X7R appears to integrate a cascade of immune responses that amplify systemic inflammation and alter central nervous system functioning, thereby linking peripheral immune dysregulation to the onset of anxiety and depression.

### 4.4. The Potential Role of the P2RX7 Gene in the Pathogenesis of Depression and Anxiety

The *P2RX7* gene, which encodes the P2X7 receptor, is located at chromosome 12q24.31 [109] in humans, an important region for MDD, bipolar, and anxiety disorders, suggesting a genetic overlap in this group of mood disorders [110]. The *P2RX7* gene comprises 13 exons and encodes the 595-amino acid protein subunit. The P2X7 receptor has 10 splice variants with diverse downstream signalling properties, and only 3 of which have been reported to be expressed in humans [111,112]. A further variation in P2X7 receptor function in humans is underlined by variation in the gene contributing to the gain or loss of function [62].

Despite the growing interest in purinergic signalling and *P2RX7* in affective disorders, studies focusing on the role of variation in this gene are scarce (Table 1). Most investigations have focused on rs2230912 within *P2RX7*, a non-synonymous coding SNP inducing an amino acid exchange (Gln460Arg). As findings on the association of this polymorphism with mood disorders have been inconsistent, Czamara et al. (2018) conducted a meta-analysis of all published data available up to that date—comprising 8652 cases and 11,153 controls—as well as unpublished results from the Munich Antidepressant Response Signature (MARS) study, to investigate the association of rs2230912 with MDD and BD. The analysis revealed a significant association between the polymorphism and combined mood disorders (MDD or BD) for the allelic, dominant, and heterozygous disadvantage models [113]. This variant is associated with modified channel function, resulting in increased Ca^2+^ influx, P2X7R dimerization, and subsequent protein–protein interactions, thereby influencing P2X7R-mediated signalling. Notably, the G allele is linked to a gain-of-function phenotype for heightened IL-1β release from monocytes upon activation [114]. This suggests that similar *P2RX7* variations in microglia may also alter cytokine release, potentially leading to neuroinflammation and impacting the functional state of neural networks, thereby increasing susceptibility to mood disorders [113,115]. However, the mechanisms through which such SNPs elevate the risk of mood disorders, or conversely, how a loss of function might confer protection against them, remain not fully elucidated [27]. Haplotype studies examined the combinations of SNPs in the region and found that rs2230912 and rs1718119 (Ala348Thr) lead to a gain-of-function variant of *P2RX7*. In fact, it seems that carrying these two polymorphisms results in increased IL-1β production in response to ATP and more severe depression [35]. Recent dimensional analyses identified significant associations among three *P2RX7* polymorphisms—the above mentioned rs2230912 (Gln460Arg), rs1718119 (Ala348Thr), and the non-coding variant rs1653625—and depressive symptomatology in two independent cohorts comprising patients with MDD and individuals with diabetes, who are at increased risk for developing mood disorders. Notably, the minor-allele haplotype rs1718119-A~rs2230912-G~rs1653625-A was associated with significantly higher depressive symptom scores compared to the most frequent G~A~C haplotype in both patient populations. These variants have also been associated with enhanced P2X7R function in prior studies as well as in the present in vitro experiments [116]. Furthermore, the two non-synonymous polymorphisms (rs2230912 and rs1718119) have previously been implicated in mood instability and emotional dysregulation, including rapid cycling in bipolar disorder (BD) [117,118]. The rs1718119-A~rs2230912-G haplotype—known to increase P2X7 pore activity [119]—has been specifically linked to rapid cycling in BD. Supporting this, a sex-specific genetic association study demonstrated that these gain-of-function alleles significantly increase risk for BD in females [120].

Our research group had previously examined several hundred variants along the *P2RX7* gene in interaction with childhood adversities on current anxiety and depressive symptoms, as well as on current suicide risk markers, employing a clumping method based on linkage disequilibrium. We found that, while *P2RX7* variation has no direct main effect on anxiety or depressive symptoms, it did moderate the effects of early-childhood traumas on these symptoms. We identified one clump of variants significantly interacting with early-childhood traumas on current anxiety levels with rs67881993 as the lead SNP, and another one in the case of depression, with rs74892325 as the lead SNP. In the case of both top SNPs, the presence of the minor allele was associated with significantly decreased BSI-Anxiety and BSI-Depression scores if the subjects experienced more severe childhood traumas, indicating a protective effect [121,122]. The finding that *P2RX7* variation interacts with early-childhood traumas to influence current anxiety levels is not only consistent with the findings of the above-discussed animal models investigating the developmental and long-term behavioural consequences of ELS and neuroinflammation but may suggest the involvement of altered P2X7 signalling in the process. Remaining at enduring consequences, variation along the gene not only had significant main effect on suicidal ideation but also altered the effects of early-childhood maltreatment on both suicidal ideation and hopelessness, a strong independent predictor of suicide risk. A clump with top SNP psy_rs11615992 was identified for the first marker and another one with index SNP rs78473339 for hopelessness. In cases of both lead SNPs, the minor alleles had a protective effect. These results suggest that *P2RX7* variation may also mediate the effect of early-childhood adversities and traumas on the later emergence of suicide risk [123].

A study of 179 Caucasian patients with anxiety disorder and syndromal panic attacks and 462 control subjects indicated a trend of association with an exonic SNP in *P2RX7* (rs1718119), with severity scores on the panic and agoraphobia scale. Patients with homozygotic AA allele scored higher in the severity of personal disability and “worries” about health versus the ones with homozygotic TT or GG alleles, suggesting a higher symptomatic load in AA allele carriers [110]. Although the P2X7 receptor rs2230912 Gln460Arg polymorphism did not present any relation to mood disorders in a case–control analysis, this receptor induces higher symptomatic severity scale scores of patients with the G allele [124]. In a cohort study, the same SNP was associated with a higher risk of developing mood disorders, anxiety, and alcoholism. This study also identified the rs208294 His155Tyr polymorphism as a possible risk factor for disease development [125]. In addition, the P2X7 receptor variant rs208294 has been associated with neuroticism-mediated outcomes of mood disorders, a personality trait that indicates vulnerability to the onset of anxiety in stressful situations [126]. Furthermore, the results suggesting that *P2RX7* variation has an effect on depression only in interaction with stress is in alignment with rodent models that explore humanized *hP2rx7* mice in a gene–environment interaction scenario. These studies report that rs2230912 heterozygous mice, when exposed to chronic social defeat, exhibit increased anhedonia and anxiety compared to homozygotes [62], leading to the conclusion that *P2RX7* specifically contributes to stress-related pathologies [127]. Mice exposed to chronic unpredictable stress or chronic restraint stress showed enhanced *P2RX7* expression in the frontal cortex and hippocampus [128,129], while the absence of the gene caused increased stress resilience and a phenotype with alleviated depression-like characteristics [30].

**Table 1 ijms-26-04687-t001:** Summary of frequently investigated *P2RX7* SNPs associated with mood disorders.

*Variant ID (SNP)*	*Associated Condition*	*Effect of Function*	*Notes*	*References*
** *rs2230912* **	MDD BD Anxiety	Gain-of-function	One of the most studied SNPs; associated with increased inflammatory response, increased pore activity of P2X7, increased receptor function	[113,124,130,131,132]
** *rs1718119* **	MDD BD (rapid cycling)	Gain-of-function	Associated with increased cytokine release, increased pore activity of P2X7, increased receptor function	[116,131]
*rs1653625*	MDD BD	Unknown	Possible influence on gene expression through miRNA-mediated regulation	[116,132]
*rs3751143*	MDD BD	Loss-of-function	Diminished receptor activity and associated with altered immune function	[116,133]
*rs7958311*	MDD	Gain-of-function and Loss-of-function	Enhances the P2X7 receptor’s channel activity, impairs the receptor’s ability to form large pores (a critical function for processes such as cytokine release and cell death)	[134]
*rs208294*	MD BD	Gain-of- function	Increases both channel and pore functions of the P2X7 receptor	[125,126]
*rs67881993*	Anxiety	Unknown	Modulating anxiety symptoms through gene–environment interactions conveying a protective effect against increased anxiety in individuals exposed to ELA	[122]
*rs74892325*	MDD	Unknown	Modulating the impact of environmental stressors on mood regulation; the presence of the minor allele was associated with a protective effect against increased depression severity in individuals exposed to ELA	[121]
*rs11615992*	Suicide risk	Regulatory role variant	Potential role in modulating the effects of early-childhood maltreatment on suicidal ideation (protective effect)	[123]
*rs78473339*	Suicide risk	Unknown	Potential role in modulating the effects of early-childhood maltreatment on hopelessness (protective effect)	[123]

MDD: major depressive disorder, BD: bipolar disorder.

## 5. P2X7R Modulation and Its Therapeutic Potential in Anxiety and Depression

In animal studies, *P2RX7* KO-mice produced a controversial relationship with anxiety-like behaviour [21,135,136,137], with no consensus on its role. The knockout of these receptors did not prevent the appearance of anxiety, while the antagonism of the receptors showed both pro-anxiety and anti-anxiety effects in different animal models. In humans, increased P2X7 receptor expression has been reported in peripheral blood mononuclear cells in anxiety disorders [138]. Inhibiting P2X7R channel activation has been shown to reduce microglial cell activation and inflammation, suggesting that P2X7R plays a role in mitigating neuronal death and neurodegenerative processes. Consequently, various drugs targeting the P2X7R channel have been developed. As a key player in initiating the inflammatory signalling cascade, P2X7R has become a potential target for antidepressant drug research [4]. Although swiftly advancing insights underscore the regulatory role of purinergic signalling on mood-related behaviour and stress reactivity, along with their pathological modifications using novel brain-penetrant P2X7R antagonists (JNJ-54175446, JNJ-55308942) undergoing phase 2 clinical trials for depression treatment [27,129], there remains an insufficient comprehension of the role of variation in *P2RX7* in the background of depressive-like behaviours.

## 6. Conclusions

Purinergic receptors are pivotal upstream regulators of the immune response; thus, the potential role of the P2X7R—the most widely studied P2 receptor—in mood disorders and anxiety has been addressed by several authors. However, only a small proportion of these studies investigated the involvement of early-childhood traumas. Synthesising findings from animal and human studies, it is evident that the P2X7 receptor and gene play a significant role in modulating the impact of stress on anxiety and depression via profound effects on several processes interacting with environmental factors across the life span. Overall, exploring the role of P2X7 receptors in mental disorders, in relation to early adversities, may not only offer a deeper understanding of the pathomechanism of various neuropsychiatric illnesses, but also a potential target for prevention and intervention. In this paper, we presented an overview of the involvement of P2X7R in anxiety and depressive disorders, focusing on the contribution of early traumas, and elucidated deficiencies in current knowledge, thereby emphasizing the need for future investigations. Future research should explore the potency of P2X7R-targeted interventions in preventing the development of anxiety and depression in individuals who have experienced early-life trauma and explore combined pharmacological and behavioural interventions. Understanding the interplay between P2X7R and early adversity is crucial, as it could pave the way for personalized interventions tailored to individuals with a history of trauma.

## Figures and Tables

**Figure 1 ijms-26-04687-f001:**
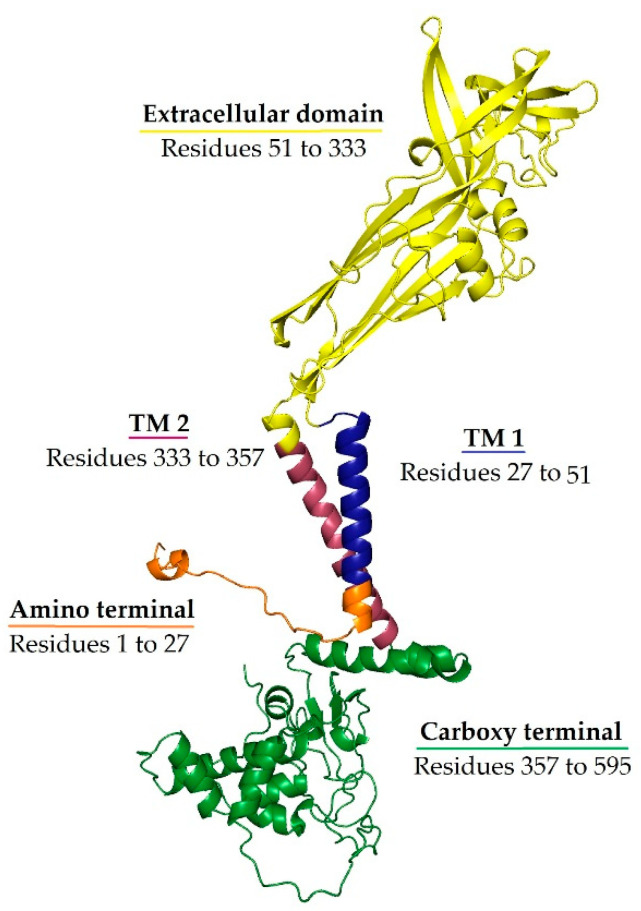
Ribbon representation of the P2X7 subunit structure. The extracellular ATP-binding domain is shown in yellow, the transmembrane domain 1 (TM1) is shown in blue, the transmembrane domain 2 (TM2) is shown in red, the intracellular amino terminal is shown in orange, and the carboxy terminal is shown in green. TM2 lines the ion-conducting pore, while the extended C-terminal domain is unique to P2X7R and is implicated in downstream signalling. Conformational changes upon ATP binding to the extracellular domain—one per subunit interface—open the channel, allowing cation flux and contributing to P2X7R’s role in neuroinflammation.

**Figure 2 ijms-26-04687-f002:**
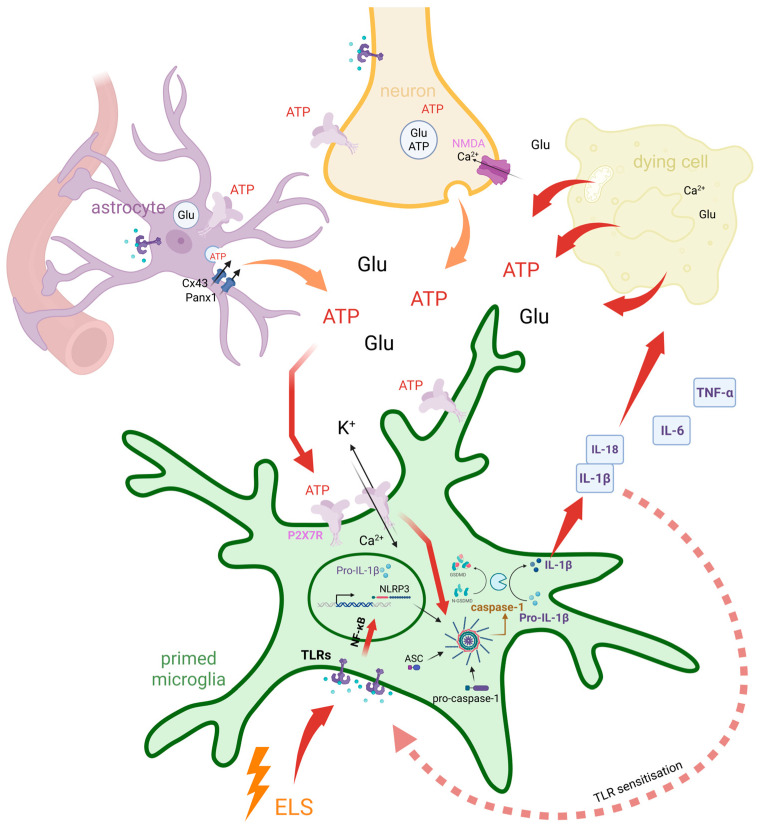
Dual feedback loops involving early-life stress and ATP-driven P2X7R signalling mediate chronic microglial neuroinflammation. Early-life stress (ELS) primes microglia by sensitizing Toll-like receptor (TLR) signalling pathways, enhancing NF-κB-mediated transcription of pro-inflammatory cytokine precursors such as pro-IL-1β. A positive feedback loop is established as IL-1β, once activated, further sensitizes TLR signalling and amplifies microglial reactivity. In parallel, extracellular ATP and glutamate (Glu) are released from neurons, astrocytes (via Cx43 and Panx1 hemichannels), and dying cells. While these molecules normally act as local messengers under physiological conditions to support homeostasis and intercellular communication, their excessive release under pathological or stress conditions contributes to microglial activation. High extracellular ATP activates P2X7Rs on microglia, leading to K^+^ efflux, Ca^2+^ influx, and subsequent NLRP3 inflammasome assembly. Caspase-1 then cleaves pro-IL-1β into mature IL-1β, which is released along with other cytokines (IL-18, IL-6, TNF-α), contributing to a second feedback loop involving cytokine signalling, ROS production, and further ATP release due to sustained inflammatory stress. These two interdependent feedback loops—one centred on ELS-induced TLR–NF-κB signalling, and the other on ATP–P2X7R–inflammasome activation—converge at key nodes including pro-IL-1β production and NLRP3 inflammasome activation. Their convergence amplifies and sustains a chronic inflammatory state, contributing to impaired neuroplasticity and long-term vulnerability to psychiatric disorders.

**Figure 3 ijms-26-04687-f003:**
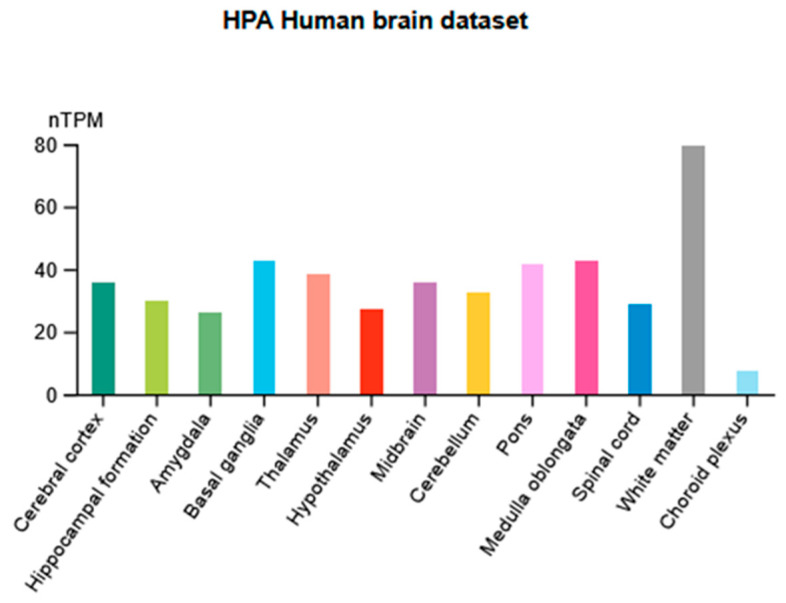
Distribution of normalized mRNA expression levels (normalized tag per million, nTPM) in 13 categories of brain regions.

**Figure 4 ijms-26-04687-f004:**
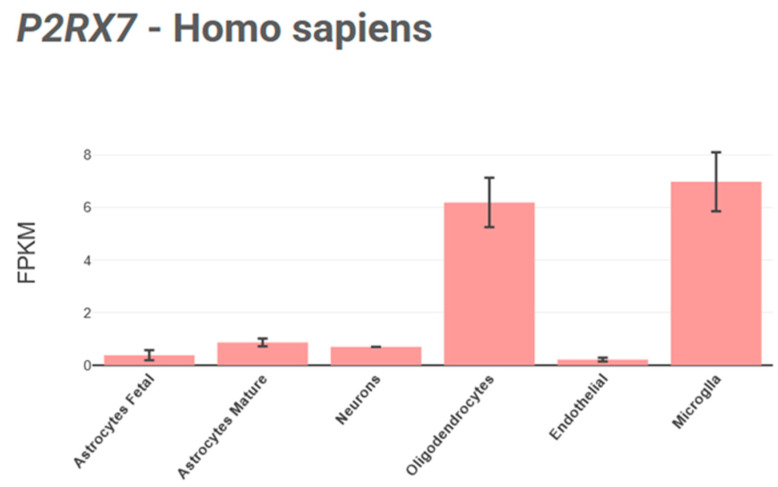
Visualization of *P2RX7* gene’s expression across human CNS cell types using the BrainRNAseq [92] database.

**Figure 5 ijms-26-04687-f005:**
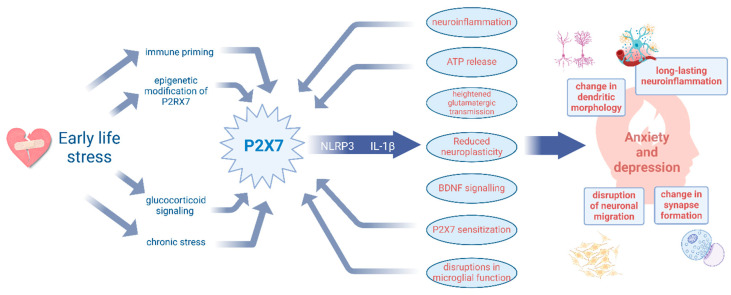
Early-life stress programs vulnerability to anxiety and depression through P2X7R-driven neuroinflammation and impaired neuroplasticity. Early-life stress (ELS) influences the P2X7R expression via immune priming, glucocorticoid signalling, chronic stress, and epigenetic modifications. P2X7R activation drives neuroinflammation via the NLRP3 inflammasome and IL-1β release, leading to ATP release, heightened glutamatergic transmission, reduced neuroplasticity, and microglial dysfunction. These downstream effects contribute to structural and functional neuronal changes, ultimately increasing susceptibility to anxiety and depression. Additionally, the backward arrows highlight potential feedback loops, where neuroinflammation with microglia activation and excessive ATP release further sensitize P2X7R signalling, fuelling the cycle of stress-induced vulnerability.

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
