# Peer review of "From Childhood Woes to Adult Blues: Unmasking the Role of Early Traumas, P2X7 Receptor, and Neuroinflammation in Anxiety and Depression"

_ijms, 2025, doi:10.3390/ijms26104687_

Round 1
Reviewer 1 Report
Comments and Suggestions for Authors
Summary
P2X7 (P2RX7) is a purinergic receptor with a well-documented role in ATP metabolism and emerging evidence connecting it to tumor progression, muscular and neurodegenerative disorders, and psychiatric conditions such as anxiety and depression. The subject of this manuscript is timely and relevant, with strong translational implications. However, the current version lacks depth in several key aspects, including structural analysis, genetic variation, and pathway integration. A more comprehensive exploration of these areas, supported by figures, bioinformatics resources, and relevant literature, is essential for improving the manuscript’s impact and scientific robustness.
Abstract
- Lines 23–26: The aim of the study should be clearly articulated, integrating specific findings that link P2RX7—both directly and indirectly—to anxiety and depressive disorders.
- Clarify the novelty of the work. Emphasize how the findings add to current understanding of P2RX7 function or pathology.
Introduction
- Provide a broader background on all P2X receptors, summarizing their structural motifs, transmembrane architecture, and calcium signaling capacity.
- The structural-functional correlation of these ion channels is a key point and should be discussed as part of the channelopathy spectrum. I recommend including these references:
- https://doi.org/10.3390/ijms150813344
- https://doi.org/10.3390/cimb46020073
- https://doi.org/10.1080/15548627.2021.1937897
- Emphasize the pathological roles of P2X7 in multiple disease systems, making the bridge to psychiatric disorders more convincing.
Manuscript Body
Protein Structure Visualization
- A figure illustrating the functional domains of P2RX7 is essential. This should include: The transmembrane domains (TM1 and TM2); The extracellular ATP-binding domain; The C-terminal domain involved in protein-protein interactions
- I suggest incorporating the AlphaFold model annotated with domain labels and functional regions.
- This visual would help correlate specific polymorphisms or mutations with their structural impact.
Subsection 3.1
The section is concise and well written. Nevertheless, I recommend adding a schematic Figure summarizing the P2RX7–ATP interaction and downstream signaling, inspired by: https://doi.org/10.1523/ENEURO.0092-22.2022; https://doi.org/10.1002/jex2.155
Line 195–197: Avoid vague statements like “is implicated.” Specify how P2RX7 contributes to depression/anxiety, perhaps via microglial activation, IL-1β production, or glutamatergic modulation.
Pathway Analysis
Line 222: Mention specific pathways as defined in GO and KEGG databases. According to the QuickGO annotations (https://www.ebi.ac.uk/QuickGO/annotations?geneProductId=Q99572), P2RX7 participates in several crucial pathways.
Subsection 3.3 – Brain Expression Patterns
- The manuscript lacks a comprehensive evaluation of P2RX7 expression in brain regions. This is essential for any neuropsychiatric interpretation.
- I suggest integrating data from: BrainSPAN – developmental expression; BrainRNAseq – cell type-specific profiles (e.g., microglia vs. neurons); Human Protein Atlas (HPA) – protein-level validation
- Add a heatmap or bar graph showing region-specific expression.
Subsection 4.4 – Genetic Contributions
- This section underrepresents the genetic landscape of P2RX7.
- Provide a summary of known SNPs and mutations associated with psychiatric conditions, citing: https://doi.org/10.1016/j.pnpbp.2019.01.006; https://doi.org/10.1016/j.pnpbp.2017.11.003
- Consider creating a table listing: Variant ID (e.g., rs3751143), Associated condition (e.g., major depression, bipolar disorder), Functional effect (e.g., gain/loss of function), Reference
- Supplement this with insights from HGMD or dbSNP.
Reviewer 2 Report
Comments and Suggestions for Authors
This manuscript reviews the potential role of the P2X7 receptor in neuroinflammation and how that may intersect with responses to early-life adverse events to drive changes in depression, anxiety, and related disorders. The manuscript is well written and concise. The topic is timely. I just have a couple minor suggestions to note.
- Sections 3.2-3.4: Inflammation and neuro-inflammation are portrayed as processes that have negative effects on the brain, but it should be remembered that inflammation is also necessary and eliminating it is also undesirable. Is there evidence that P2X7 receptor signaling is also a positive, in at least some circumstances?
- Conclusion, lines 362-363: The statement that P2X7 “mediates the impact of stress on anxiety and depression” I think overstates the case that can be made on the data available so far. It is clear that this signaling pathway is important, but saying that it “mediates the effects” implies that this is the most important pathway. I suggest more caution in the wording here.
Round 2
Reviewer 1 Report
Comments and Suggestions for Authors
Authors addressed all the reviewer's comments